# MOLE-BERT: RETHINKING PRE-TRAINING GRAPH NEURAL NETWORKS FOR MOLECULES

**Jun Xia**[1†]**, Chengshuai Zhao**[1,2†]**, Bozhen Hu**[1]**, Zhangyang Gao**[1]**,
Cheng Tan**[1]**, Yue Liu**[1]**, Siyuan Li**[1]**, Stan Z. Li**[1*]
[1]AI Lab, Research Center for Industries of the Future, Westlake University
[2]University of California, Irvine
{xiajun, Stan.ZQ.Li}@westlake.edu.cn; chengsz4@uci.edu

## ABSTRACT

Recent years have witnessed the prosperity of pre-training graph neural networks (GNNs) for molecules. Typically, atom types as node attributes are randomly masked, and GNNs are then trained to predict masked types as in AttrMask (Hu et al., 2020), following the Masked Language Modeling (MLM) task of BERT (Devlin et al., 2019). However, unlike MLM with a large vocabulary, the AttrMask pre-training does not learn informative molecular representations due to small and unbalanced atom 'vocabulary'. To amend this problem, we propose a variant of VQ-VAE (Van Den Oord et al., 2017) as a context-aware tokenizer to encode atom attributes into chemically meaningful discrete codes. This can enlarge the atom vocabulary size and mitigate the quantitative divergence between dominant (e.g., carbons) and rare atoms (e.g., phosphorus). With the enlarged atom 'vocabulary', we propose a novel node-level pre-training task, dubbed Masked Atoms Modeling (**MAM**), to mask some discrete codes randomly and then pre-train GNNs to predict them. MAM also mitigates another issue of AttrMask, namely the negative transfer. It can be easily combined with various pre-training tasks to improve their performance. Furthermore, we propose triplet masked contrastive learning (**TMCL**) for graph-level pre-training to model the heterogeneous semantic similarity between molecules for effective molecule retrieval. MAM and TMCL constitute a novel pre-training framework, **Mole-BERT**, which can match or outperform state-of-the-art methods in a fully data-driven manner. We release the code at https://github.com/junxia97/Mole-BERT.

## 1 INTRODUCTION

Pre-training Language Models (PLMs) have revolutionized the landscape of Natural Language Processing (NLP) (Qiu et al., 2020b; Zheng et al., 2022). The representative one is BERT (Devlin et al., 2019), whose Masked Language Modeling (MLM) task first randomly masks some proportions of tokens within a text, and then recovers the masked tokens based on the encoding results of the corrupted text. Although BERT also includes the pre-training task of next sentence prediction, MLM is verified as the only success recipe for BERT (Liu et al., 2019). Inspired by this, MLM-style pre-training task has been extended to many other domains (Hu et al., 2020; He et al., 2022).

Molecules can be naturally represented as graphs with their atoms as nodes and chemical bonds as edges. Hence, Graph Neural Networks (GNNs) can be utilized to process molecular graph data. To exploit the abundant unlabeled molecules, tremendous efforts have been devoted to pre-training GNNs for molecular graph representations (Xia et al., 2022e). The pioneering work (Hu et al., 2020) on this topic first pre-trains GNNs with a MLM-style pre-training task (AttrMask) on large-scale unlabeled molecular graph datasets. Specifically, they randomly mask some proportions of atoms and then pre-train the models to predict them. AttrMask has emerged as a fundamental pre-training task and many subsequent works adopt it as a sub-task for pre-training (Zhang et al., 2021; Li et al., 2021a). During the tuning stage, researchers replace the top layer of the pre-trained models with a task-specific sub-network and train the new model with the labeled molecules of the downstream tasks. However,

---

[†]Equal Contribution, [*]Corresponding Author

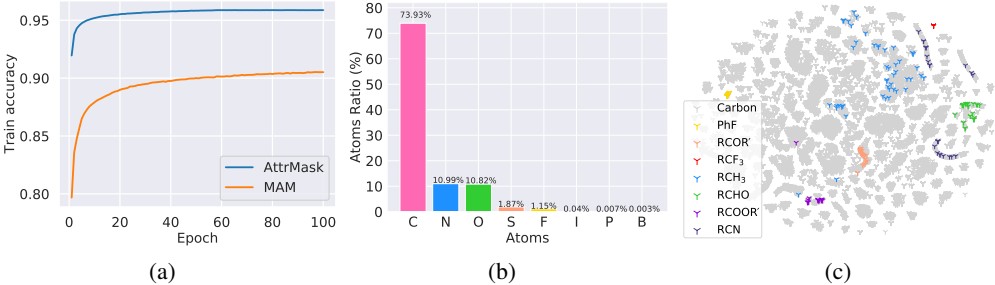

Figure 1: **(a)**: Pre-training accuracy curves of AttrMask and MAM; **(b)**: The atoms ratios of various chemical elements in the pre-training datasets; **(c)**: The t-SNE visualization of the carbon representations learned by the proposed tokenizer. Firstly, we randomly sample 30, 000 carbons from the QM9 dataset (Ruddigkeit et al., 2012). Among them, 250 are randomly chosen and colored based on the types of functional groups that carbons belong to. Both R and R' are the abbreviations for groups in the rest of a molecule. The details of the 7 local structure groups are listed in Appendix I.

Hu et al. (2020) observe that pre-training only with AttrMask (node-level pre-training task) will incur the negative transfer issue (i.e., pre-trained models fall behind no pre-trained models) sometimes. Intuitively, they think this phenomenon can be attributed to the lack of graph-level pre-training tasks and thus introduce supervised graph-level pre-training strategies, which are impractical because the labels are often expensive or unavailable. Additionally, some supervised pre-training tasks unrelated to the downstream task of interest can even degrade the downstream performance (Hu et al., 2020).

In this paper, we provide a second voice on this predominant belief and aim to explain the negative transfer in molecular graph pre-training. Firstly, as can be observed in Figure 1(a), the pre-training accuracy of AttrMask converges to $\sim 96\%$ quickly, which indicates AttrMask task (118-way classification, 118 is the number of common chemical elements in nature) is extremely simple for the small atom vocabulary size. The atom vocabulary is the set of unique atom types of common chemical elements in nature. In contrast, the training accuracy of the MLM task ($\sim 30k$-way classification) in BERT only grows to 70% and hardly converges for the large text vocabulary size ($\sim 30k$ tokens) (Kosec et al., 2021). Text vocabulary is the set of unique tokens in the corpus. Secondly, the quantitative divergence between different atoms is extremely significant (see Figure 1(b)), which will bias the models' prediction toward dominant atoms (e.g., carbons) and lead to fast convergence. Previous works (Clark et al., 2020; Robinson et al., 2021) have revealed that simple pre-training tasks will capture less transferable knowledge and impair the generalization or adaptation to novel tasks.

Tokenization is the first step in any NLP pipeline, which separates a piece of text into smaller units called tokens (Sennrich et al., 2015). Hence, pre-training language models include two stages: the first stage is tokenizer training and the second stage is language models pre-training. However, for GNNs pre-training, previous works adopt the atoms' types as tokens, which will result in a small-size and unbalanced atom vocabulary. We argue that atoms with different contexts should be tokenized into different discrete values even if they belong to the same type. For example, aldehyde carbons and ester carbons indicate different properties of molecules even if both of them are carbons. Hence, we introduce a context-aware tokenizer to encode atoms to meaningful discrete values. Specifically, these discrete values are the latent codes of a variant of graph VQ-VAE (Van Den Oord et al., 2017). The tokenizer is context-aware because the encoder of graph VQ-VAE is a GNN model. In this way, we can categorize the dominant atoms (e.g., carbons) into several chemically meaningful sub-classes (e.g., aldehyde carbons, ester carbons, etc.) considering the atoms' contexts, which will enlarge the atom vocabulary size and mitigate the quantitative divergence between dominant and rare atoms. To support the above claims, we provide the t-SNE visualization of carbon representations learned by the proposed tokenizer in Figure 1(c). As can be observed, the representations of carbons are clustered based on types of functional groups, which indicates our tokenizer can encode atoms to chemically meaningful values. With the new vocabulary, we propose a node-level pre-training task, dubbed Masked Atoms Modeling (MAM), to randomly mask the discrete values and pre-train GNNs to predict them. For molecular graph-level pre-training, graph contrastive learning (You et al., 2020) is a feasible pre-training strategy. However, contrastive approaches push different molecules away equally regardless of their true degrees of similarities (Xia et al., 2022c; Liu et al., 2023; 2022c). To

remedy this deficiency, we propose triplet masked contrastive learning (TMCL), which mimics the various degrees of molecular similarities with different masking ratios.

We highlight the following contributions: **(i)** We find the negative transfer issue of AttrMask can be attributed to the extremely small and unbalanced atom vocabulary. **(ii)** As a remedy, we contribute a context-aware tokenizer for molecular graphs using a variant of VQ-VAE. Also, the tokenizer can be re-used as an off-the-shelf tool for subsequent works like the NLP community. **(iii)** With the new vocabulary, we propose a tailored pre-training task, MAM, to alleviate the negative transfer issue. MAM serves as a fundamental pre-training task and can be combined with various pre-training tasks to advance their performance. **(iv)** We propose a novel graph-level pre-training task, TMCL, to model heterogeneous similarities between molecules, which is especially effective for molecule retrieval. **(v)** We combine MAM and TMCL as a joint pre-training framework (Mole-BERT), which matches or outperforms state-of-the-art models that require expensive domain knowledge as guidance.

## 2 RELATED WORK

### 2.1 PRE-TRAINING ON MOLECULES

Neural Networks have achieved remarkable success in molecular representation learning. While effective and prevalent, they require expensive annotations and barely generalize to unseen molecules (Tan et al., 2021; Xia et al., 2021), which poses a hurdle to practical applications. To remedy these deficiencies, tremendous efforts have been devoted to pre-training on molecules. Initially, one line of these works (Wang et al., 2019; Chithrananda et al., 2020) adopts MLM-style pre-training strategy on molecular SMILES (Weininger et al., 1989) strings. Subsequently, recent works follow the contrastive paradigm (Zhu et al., 2021b;a; Qiu et al., 2020a; Liu et al., 2022b; Zheng et al., 2023). For molecular pre-training, GraphCL (You et al., 2020) and its variants (You et al., 2021; Suresh et al., 2021; Xia et al., 2022b; Wang et al., 2022; Sun et al., 2021; Fang et al., 2022b) embed augmented versions of the anchor molecular graph close to each other and push the embeddings of other molecules apart. Additionally, DGI (Velickovic et al., 2019) and InfoGraph (Sun et al., 2020a) is proposed to obtain expressive representations for graphs or nodes via maximizing the mutual information between graph-level representations and substructure-level representations of different granularity. The other line of work adopts generative or predictive pretext tasks. Typically, GPT-GNN (Hu & others., 2020) introduces an attributed graph generation task to pre-train GNNs so that they can capture the structural and semantic properties of the graph. For molecular graphs, Hu et al. (2020) and Li et al. (2021b) conduct attribute and structure prediction at the level of individual nodes as well as entire graphs. To capture the rich information in molecular graph motifs, GROVER (Rong et al., 2020a) and MGSSL (Zhang et al., 2021) propose to predict or generate the motifs. Considering that 3D geometric information plays a vital role in predicting molecular properties, several recent works (Liu et al., 2022a; Stärk et al., 2021; Fang et al., 2022a; Zhu et al., 2022) pre-train the GNN encoders on molecular datasets with 3D geometric information. We recommend readers refer to a recent survey (Xia et al., 2022f) for more relevant literature. Many above-mentioned works adopt AttrMask (Hu et al., 2020) as a fundamental pre-training sub-task. However, AttrMask will incur the negative transfer issue sometimes. We explain this phenomenon and contribute a novel pre-training strategy to remedy this deficiency.

### 2.2 MLM-STYLE PRE-TRAINING STRATEGIES

The masked language modeling (MLM) task proposed in BERT (Devlin et al., 2019) has emerged as one of the most popular and successful pre-training tasks. Empirically, RoBERTa (Liu et al., 2019) finds that MLM is the only success recipe of BERT and discards the sentence-level task in BERT. Also, BART (Lewis et al., 2019) and T5 (Raffel et al., 2020) both observed that MLM is often the most effective task. Alternatively, various MLM variants improve MLM with dynamic masking strategy (Liu et al., 2019) and blockwise masking strategy (Joshi et al., 2020). Additionally, XLNet (Yang et al., 2019) proposes permutation language modeling that conducts MLM in an autoregressive manner. Inspired by these advances, for vision transformer pre-training, the model receives incomplete images with a large portion of the patches removed and learns to reconstruct the missing contents on low-level image pixels (He et al., 2022; Li et al., 2021c; Dosovitskiy et al., 2021; Chen et al., 2020), high-level semantics (Bao et al., 2021) or hand-crafted feature descriptors (Wei et al., 2022). Different from the above-mentioned works, our MAM adopts the encoder of VQ-VAE

as the atoms tokenizer, which discretizes a continuous semantic space to discrete codes in a context-aware manner. Moreover, we observe that atoms of different types might be allocated with the same token id with vanilla VQ-VAE. As a remedy, we introduce a group VQ-VAE to address this issue.

## 3 PRELIMINARY

Graph Neural Networks (GNNs) are the dominant tools for modeling graph data (Kipf & Welling, 2016; Velickovic et al., 2018; Hamilton et al., 2017; Xu et al., 2019). The structure of graph data guides the aggregation of local neighborhood information and leads to a more contextual representation for each node. Also, we can adopt a graph pooling operation (Mesquita et al., 2020) to get the representation for the whole graph. Let $\mathcal{G} = (\mathcal{V}, \mathcal{E})$ denotes a graph with node attributes $x_v$ for $v \in \mathcal{V}$ and edge attributes $e_{uv}$ for $(u, v) \in \mathcal{E}$. Formally, supposing that $h_v^{(l)}$ is the representation of node $v$ at the $l$-th GNN layer and $\mathcal{N}(v)$ are all the neighbor nodes of node $v$, the update procedure from the $(l-1)$-th layer to the $l$-th layer is:

$$h_v^{(l)} = \text{COMBINE}\left(h_v^{(l-1)}, \text{AGGREGATE}\left(\left\{\left(h_v^{(l-1)}, h_u^{(l-1)}, e_{uv}\right) : u \in \mathcal{N}(v)\right\}\right)\right), \quad (1)$$

where $e_{uv}$ denotes the edge between node $u$ and $v$. $\text{AGGREGATE}(\cdot)$ is the aggregation function (e.g., mean operator) of the neighborhood information. $\text{COMBINE}(\cdot)$ combines the information of neighbours and node $v$ (e.g., concatenation operator). After $L$ iterations of message passing, the hidden states $h_v^{(L)}$ in the last iteration are the embeddings of $v$. Finally, we adopt a $\text{READOUT}(\cdot)$ operation (e.g., averaging, sum or graph pooling) to get the representation $\mathbf{h}_G$ for the whole graph $G$:

$$\mathbf{h}_G = \text{READOUT}\left(\left\{h_v^{(L)} \mid v \in \mathcal{V}\right\}\right). \quad (2)$$

## 4 PROPOSED PRE-TRAINING FRAMEWORK: MOLE-BERT

In this section, we elaborate on the proposed pre-training framework Mole-BERT, which contains a node-level per-training approach (MAM) and a graph-level pre-training approach (TMCL).

### 4.1 MASKED ATOMS MODELING (MAM)

Similar to NLP, we represent the atoms as discrete tokens using group VQ-VAE shown in Figure 2. Formally, the atoms $\mathcal{V} = \{v_1, v_2, \cdots, v_n\}$ of a molecule graph $\mathcal{G}$ is tokenized to $\mathbf{z} = \{z_1, z_2, \cdots, z_n\} \in \mathcal{A}^n$, where the atom vocabulary $\mathcal{A}$ contain $|\mathcal{A}|$ ($|\mathcal{A}| = 512$) discrete codes. Firstly, the GNN encoder of group VQ-VAE encodes the atoms to atoms embeddings. Next, the vector quantizer (VQ) looks up the nearest neighbor in the codebook for each atom embedding $h_i$. Let $\{e_1, e_2, \cdots, e_{|\mathcal{A}|}\}$ denote the codebook embeddings. The quantized code of the $i$-th atom is:

$$z_i = \text{argmin}_j \|h_i - e_j\|_2. \quad (3)$$

After quantizing the atoms to discrete tokens, we feed the corresponding codebook embeddings $\{e_{z_1}, e_{z_2}, \cdots, e_{z_n}\}$ to the decoder to reconstruct the input molecule graph. Kindly note that the vector quantization process is non-differentiable. In order to train the encoder, the gradient is approximated like the straight-through estimator (Bengio et al., 2013) and copied from the decoder to the encoder. Intuitively, the quantizer looks up the nearest code for each encoder output, so the gradients of codebook embeddings can be utilized to train the encoder approximately. With the input attributes $v_i$ and reconstructed attributes $\widehat{v_i}$, the training loss of the tokenizer for the molecule graph $\mathcal{G}$ is,

$$\mathcal{L}_{\text{VQ}} = \frac{1}{n}\sum_{i=1}^{n}\left(1 - \frac{v_i^T \widehat{v_i}}{\|v_i\| \cdot \|\widehat{v_i}\|}\right)^{\gamma} + \frac{1}{n}\sum_{i=1}^{n}\|\text{sg}[h_i] - e_{z_i}\|_2^2 + \frac{\beta}{n}\sum_{i=1}^{n}\|\text{sg}\left[e_{z_i}\right] - h_i\|_2^2, \quad (4)$$

where the first term is a reconstruction loss with the scaled cosine error ($\gamma \geq 1$), the second term is a VQ loss aiming to update the codebook and the third term is a commitment loss which encourages the output of the encoder to stay close to the chosen codebook embedding. $\text{sg}[\cdot]$ denotes stop-gradient, $\beta$ is a hyper-parameter set to 0.25 in our experiments. This idea is inspired by DALL-E (Ramesh et al., 2021) which uses discrete VAE as a tokenizer for text-to-image generation. We show the superiority of our tokenizer over discrete VAE empirically in Appendix F and clarify the difference

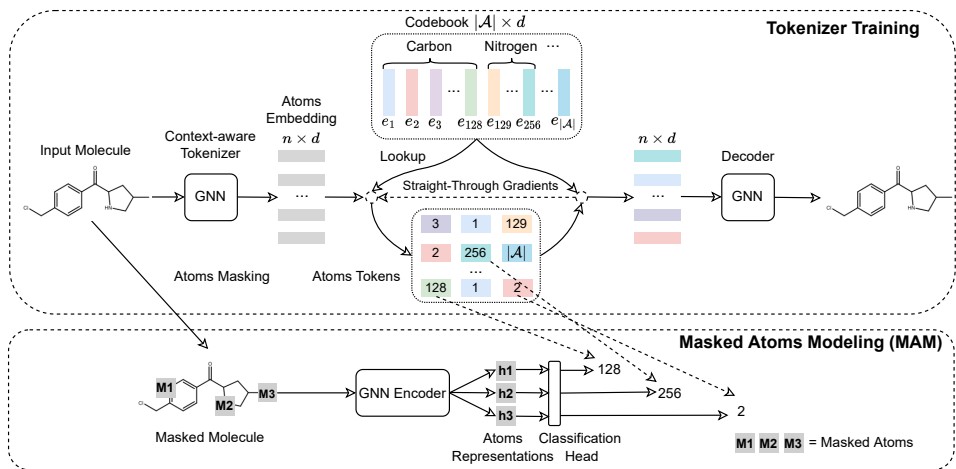

Figure 2: Schematic diagram of the tokenizer training and masked atoms modeling (MAM).

here: (1) Our tokenizer tokenizes atoms to discrete codes in a context-aware manner; (2) We use the tokens for pre-training instead of autoregressive generation; (3) We observe that atoms of different types might be allocated with the same token id with vanilla VQ-VAE. As a remedy, we introduce a group VQ-VAE to address this issue. Specifically, we divide the codebook embeddings into several groups, each of which corresponds to specific atom types. For example, the quantized codes of carbon, nitrogen and oxygen are restricted to $[1, 128]$, $[129, 256]$ and $[257, 384]$, respectively. The left rare atoms are restricted to $[385, 512]$ because they are less likely to conflict with each other. Additionally, we will release the tokenizer as an off-the-shelf tool for better or larger-scale molecular graph pre-training, just like what WordPiece (Wu et al., 2016) and BPE (Sennrich et al., 2015) have done for the NLP community. With the new tokenizer, we propose MAM to pre-train GNNs. Specifically, given an input molecule graph $\mathcal{G}$, we randomly mask its 15% atoms' tokens and pre-train GNNs to predict them. We study the influence of the masking ratios in Appendix C. We term the masked atoms' index set as $\mathcal{M}$ and the masked molecular graph as $\mathcal{G}^{\mathcal{M}}$. For each masked atom $i \in \mathcal{M}$, a softmax classifier is adopted to predict the discrete values over the vocabulary $\mathcal{A}$. The pre-training loss of MAM is:

$$\mathcal{L}_{\texttt{MAM}} = -\sum_{\mathcal{G} \in \mathcal{D}} \sum_{i \in \mathcal{M}} \log p\left(z_i \mid \mathcal{G}^{\mathcal{M}}\right),\tag{5}$$

where $\mathcal{D}$ denotes the datasets and $z_i$ is the token of the atom $v_i$.

## 4.2 GRAPH-LEVEL TASK: TRIPLET MASKED CONTRASTIVE LEARNING (TMCL)

Although MAM can mitigate the negative transfer issue, we observe that it fails to capture molecular graph-level semantics. Specifically, we first calculate the widely-used Tanimoto coefficient (Bajusz et al., 2015) of the extended connectivity fingerprints (ECFP) (Rogers & Hahn, 2010) between two molecules as their chemical similarity. Then, we pick the molecule pairs with top 15% similarity as 'similar' ones and the left 85% molecule pairs in the datasets are 'random' pairs. Figure 3(a) shows significant inconsistency between the learned representations (by MAM) and ECFP, which will impair molecule retrieval using MAM (See Section 5.3) because two random molecules may have high similarity scores (lack of uniformity), while closely related molecules may have more different representations (lack of alignment) (Wang & Isola, 2020). On the other hand, existing graph-level pre-training tasks often follow a supervised paradigm (Hu et al., 2020), where the labels for molecules are expensive for the laborious wet-lab experiments. Graph contrastive learning (You et al., 2020) is a possible remedy for the above issues. For each molecule (anchor), they maximize the agreement between paired molecular graph augmentations (positive pairs) and push away other molecules in the batch as negative pairs (dissimilar molecules) indiscriminately. However, we argue that this framework cannot reflect the heterogeneous similarities between the anchor and other molecules. For example, the similarity between formic acid (negative) and acetic acid (anchor) should

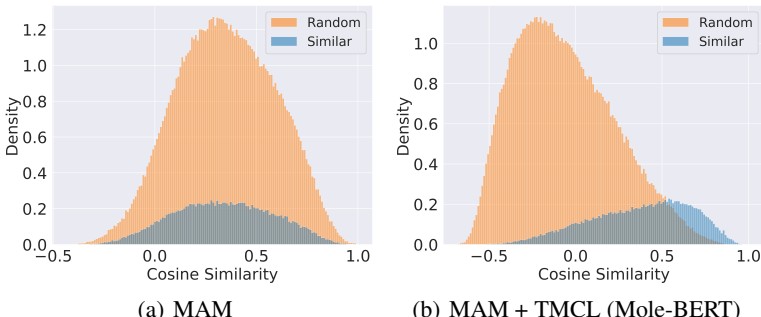

Figure 3: Similarity histograms of MAM and Mole-BERT on Toxcast dataset. Cosine similarity measures the similarity between the learned representations while 'Random' and 'Similar' are defined by the chemical similarity between molecular fingerprints.

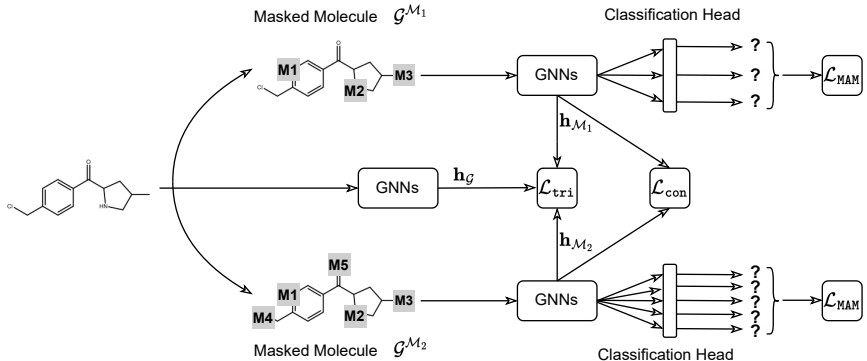

Figure 4: Schematic diagram of Mole-BERT framework. The 3 GNNs here share the same encoder.

be more significant than the one between ethanol (negative) and acetic acid (anchor). Hence, we introduce triplet masked contrastive learning, dubbed TMCL, to mitigate this crucial defect. More specifically, for each molecular graph $\mathcal{G}$, we first generate its augmented version $\mathcal{G}^{\mathcal{M}_1}$ with masked atoms index $\mathcal{M}_1$ and a smaller masking ratio (e.g., 15%). Then we enlarge the masking ratio (e.g., 30%) and obtain the other augmented version $\mathcal{G}^{\mathcal{M}_2}$ with atoms index $\mathcal{M}_2$. Now, we constitute a triplet $(\mathcal{G}, \mathcal{G}^{\mathcal{M}_1}, \mathcal{G}^{\mathcal{M}_2})$ with latent relation among them, i.e., $\mathcal{G}^{\mathcal{M}_1}$ is more similar to $\mathcal{G}$ than $\mathcal{G}^{\mathcal{M}_2}$ to $\mathcal{G}$. Although index $\mathcal{M}_1$ sometimes lies in some crucial atoms (e.g., functional group) that are indicative of properties, the latent relation holds true in most cases and pre-training with abundant data will help alleviate this issue. Given that $\mathbf{h}_\mathcal{G}, \mathbf{h}_{\mathcal{M}_1}, \mathbf{h}_{\mathcal{M}_2}$ are the graph-level representations for $\mathcal{G}, \mathcal{G}^{\mathcal{M}_1}$ and $\mathcal{G}^{\mathcal{M}_2}$, respectively, we can model such latent relation with the triplet loss,

$$\mathcal{L}_{\texttt{tri}} = \sum_{\mathcal{G} \in \mathcal{D}} \max\left(sim\left(\mathbf{h}_\mathcal{G}, \mathbf{h}_{\mathcal{M}_2}\right) - sim\left(\mathbf{h}_\mathcal{G}, \mathbf{h}_{\mathcal{M}_1}\right), 0\right), \quad (6)$$

where $sim(\cdot, \cdot)$ denotes the cosine similarity. Now, we can combine the triplet loss $\mathcal{L}_{\texttt{tri}}$ with commonly-used contrastive loss $\mathcal{L}_{\texttt{con}}$ as graph-level pre-training objective $\mathcal{L}_{\texttt{TMCL}}$,

$$\mathcal{L}_{\texttt{TMCL}} = \mathcal{L}_{\texttt{con}} + \mu\mathcal{L}_{\texttt{tri}}, \quad \text{where} \quad \mathcal{L}_{\texttt{con}} = -\sum_{\mathcal{G} \in \mathcal{D}} \log \frac{e^{sim(\mathbf{h}_{\mathcal{M}_1}, \mathbf{h}_{\mathcal{M}_2})/\tau}}{\sum_{\mathcal{G}' \in \mathcal{B}} e^{sim(\mathbf{h}_{\mathcal{M}_1}, \mathbf{h}_{\mathcal{G}'})/\tau}}, \quad (7)$$

where $\mathcal{B}$ is the sampled batch including $\mathcal{G}$, $\mu$ is the trade-off hyperparameter, and $\tau$ is the temperature hyper-parameter. Finally, MAM and TMCL constitute a unified pre-training framework, Mole-BERT (see Figure 4), whose hybrid loss is,

$$\mathcal{L}_{\texttt{Mole-BERT}} = \mathcal{L}_{\texttt{MAM}} + \mathcal{L}_{\texttt{TMCL}}. \quad (8)$$

# 5 EXPERIMENTS

## 5.1 DATASETS

For the pre-training stage, we use 2 million molecules sampled from the ZINC15 database (Sterling & Irwin, 2015) following previous works (Hu et al., 2020). The main downstream task is molecular property prediction, where we adopt the widely-used 8 binary classification datasets contained in MoleculeNet (Wu et al., 2018). Kindly note that we use *scaffold splitting* (Ramsundar et al., 2019), which splits the molecules according to their structures to mimic real-world use cases. Additionally, we validate the effectiveness of Mole-BERT on a broader range of downstream tasks and datasets (See Section 5.3). The detailed information of all the datasets can be seen in Appendix A.

## 5.2 EXPERIMENTS CONFIGURATION

We use a 5-layer Graph Isomorphism Networks (GINs) whose hidden dimension is 300 (Xu et al., 2019) as the backbone architecture, which is one of the state-of-the-art GNNs for graph-level tasks. We adopt mean pooling as the readout function. During the pre-training stage, GNNs are pre-trained for 100 epochs with batch-size as 256 and the learning rate as 0.001. During the fine-tuning stage, we train for 100 epochs with batch-size as 32 and report the test score with the best cross-validation performance. The split for train/validation/test sets is $80\% : 10\% : 10\%$. The hyper-parameter $\mu$ is picked from $\{0.1, 0.3, 0.5\}$ with the validation set. Additionally, considering that previous works adopt different evaluation protocols, we reproduce all the results with the same protocol as the pioneering work (Hu et al., 2020) rigorously for fairness. Hence, the results of some baselines may differ from their original papers. *More details can be found in Appendix B and Appendix E.*

Table 1: Results for molecular property prediction (classification). We report the mean (standard deviation) ROC-AUC of 10 random seeds with scaffold splitting. The best results and the second best are highlighted with **bold** and bold, respectively. 'No pre-train' means training from scratch.

| | Tox21 | ToxCast | Sider | ClinTox | MUV | HIV | BBBP | Bace | Average |
|---|---|---|---|---|---|---|---|---|---|
| # Molecules | 7,831 | 8,575 | 1,427 | 1,478 | 93,087 | 41,127 | 2,039 | 1,513 | - |
| No pretrain | 74.6 (0.4) | 61.7 (0.5) | 58.2 (1.7) | 58.4 (6.4) | 70.7 (1.8) | 75.5 (0.8) | 65.7 (3.3) | 72.4 (3.8) | 67.15 |
| InfoGraph (Sun et al., 2020b) | 73.3 (0.6) | 61.8 (0.4) | 58.7 (0.6) | 75.4 (4.3) | 74.4 (1.8) | 74.2 (0.9) | 68.7 (0.6) | 74.3 (2.6) | 70.10 |
| GPT-GNN (Hu & others., 2020) | 74.9 (0.3) | 62.5 (0.4) | 58.1 (0.3) | 58.3 (5.2) | 75.9 (2.3) | 65.2 (2.1) | 64.5 (1.4) | 77.9 (3.2) | 68.45 |
| EdgePred (Hamilton et al., 2017) | 76.0 (0.6) | 64.1 (0.6) | 60.4 (0.7) | 64.1 (3.7) | 75.1 (1.2) | 76.3 (1.0) | 67.3 (2.4) | 77.3 (3.5) | 70.08 |
| ContextPred (Hu et al., 2020) | 73.6 (0.3) | 62.6 (0.6) | 59.7 (1.8) | 74.0 (3.4) | 72.5 (1.5) | 75.6 (1.0) | 70.6 (1.5) | 78.8 (1.2) | 70.93 |
| GraphLoG (Xu et al., 2021a) | 75.0 (0.6) | 63.4 (0.6) | 59.6 (1.9) | 75.7 (2.4) | 75.5 (1.6) | 76.1 (0.8) | 68.7 (1.6) | 78.6 (1.0) | 71.56 |
| G-Contextual (Rong et al., 2020b) | 75.0 (0.6) | 62.8 (0.7) | 58.7 (1.0) | 60.6 (5.2) | 72.1 (0.7) | 76.3 (1.5) | 69.9 (2.1) | 79.3 (1.1) | 69.34 |
| G-Motif (Rong et al., 2020b) | 73.6 (0.7) | 62.3 (0.6) | 61.0 (1.5) | 77.7 (2.7) | 73.0 (1.8) | 73.8 (1.2) | 66.9 (3.1) | 73.0 (3.3) | 70.16 |
| AD-GCL (Suresh et al., 2021) | 74.9 (0.4) | 63.4 (0.7) | 61.5 (0.9) | 77.2 (2.7) | 76.3 (1.4) | 76.7 (1.2) | 70.7 (0.3) | 76.6 (1.5) | 72.16 |
| JOAO (You et al., 2021) | 74.8 (0.6) | 62.8 (0.7) | 60.4 (1.5) | 66.6 (3.1) | 76.6 (1.7) | 76.9 (0.7) | 66.4 (1.0) | 73.2 (1.6) | 69.71 |
| SimGRACE (Xia et al., 2022b) | 74.4 (0.3) | 62.6 (0.7) | 60.2 (0.9) | 75.5 (2.0) | 75.4 (1.3) | 75.0 (0.6) | 71.2 (1.1) | 74.9 (2.0) | 71.15 |
| GraphCL (You et al., 2020) | 75.1 (0.7) | 63.0 (0.4) | 59.8 (1.3) | 77.5 (3.8) | 76.4 (0.4) | 75.1 (0.7) | 67.8 (2.4) | 74.6 (2.1) | 71.16 |
| GraphMAE (Hou et al., 2022) | 75.2 (0.9) | 63.6 (0.3) | 60.5 (1.2) | 76.5 (3.0) | 76.4 (2.0) | 76.8 (0.6) | 71.2 (1.0) | 78.2 (1.5) | 72.30 |
| 3D InfoMax (Stärk et al., 2022) | 74.5 (0.7) | 63.5 (0.8) | 56.8 (2.1) | 62.7 (3.3) | 76.2 (1.4) | 76.1 (1.3) | 69.1 (1.2) | 78.6 (1.9) | 69.69 |
| GraphMVP (Liu et al., 2022a) | 74.9 (0.8) | 63.1 (0.2) | 60.2 (1.1) | **79.1** (2.8) | 77.7 (0.6) | 76.0 (0.1) | 70.8 (0.5) | 79.3 (1.5) | 72.64 |
| MGSSL (Zhang et al., 2021) | 75.2 (0.6) | 63.3 (0.5) | 61.6 (1.0) | 77.1 (4.5) | 77.6 (0.4) | 75.8 (0.4) | 68.8 (0.6) | 78.8 (0.9) | 72.28 |
| AttrMask (Hu et al., 2020) | 75.1 (0.9) | 63.3 (0.6) | 60.5 (0.9) | 73.5 (4.3) | 75.8 (1.0) | 75.3 (1.5) | 65.2 (1.4) | 77.8 (1.8) | 70.81 |
| MAM (with vanilla VQ-VAE) | 75.8 (0.6) | 63.1 (0.5) | 60.7 (1.5) | 74.2 (2.7) | 76.5 (1.6) | 76.2 (0.9) | 66.4 (0.7) | 78.2 (0.8) | 71.39 |
| TMCL (w/o $\mathcal{L}_{\tt con}$) | 73.5 (1.0) | 61.8 (0.3) | 58.7 (1.6) | 61.1 (4.1) | 71.6 (1.3) | 73.5 (1.3) | 65.4 (2.6) | 73.7 (2.4) | 67.41 |
| TMCL (w/o $\mathcal{L}_{\tt tri}$) | 74.1 (0.4) | 62.4 (0.8) | 58.7 (3.0) | 75.6 (2.2) | 75.7 (1.1) | 74.6 (1.1) | 66.8 (1.4) | 74.2 (1.3) | 70.26 |
| MAM | 76.2 (0.5) | 63.9 (0.3) | 61.4 (1.9) | 75.1 (3.0) | 77.4 (2.1) | 77.5 (1.0) | 66.8 (1.5) | 78.9 (1.1) | 72.16 |
| TMCL | 74.9 (0.7) | 63.2 (0.7) | 59.6 (1.4) | 77.0 (4.2) | 77.2 (0.3) | 75.3 (1.1) | 67.6 (1.3) | 75.1 (1.2) | 71.24 |
| **Mole-BERT** | **76.8** (0.5) | **64.3** (0.2) | **62.8** (1.1) | 78.9 (3.0) | **78.6** (1.8) | **78.2** (0.8) | **71.9** (1.6) | **80.8** (1.4) | **74.04** |

## 5.3 RESULTS AND ANALYSIS

**Primary Results and Analysis.** We document the main results of molecular property prediction in Table 1 and Table 2. Our systematic study suggests the following trends:

**Observation I:** The pre-training task of AttrMask incurs negative transfer issue on some datasets (HIV and BBBP). In contrast, MAM achieves consistent and notable improvements over AttrMask and 'No pretrain', although MAM pre-trains GNNs only with a node-level task. This observation verifies that only node-level pre-training tasks can also mitigate the negative transfer, which overturns the previous belief that pre-training GNNs at the level of individual nodes may give limited improvements. The reasons why AttrMask fails lie in the extremely small and unbalanced atoms vocabulary.

**Observation II:** Mole-BERT can achieve competitive or better performance than previous pre-training strategies under the same experimental protocols. More specifically, Mole-BERT outperforms 'No pretrain' model by 6.89% and the current state-of-the-art method GraphMVP by nearly 1.40%, though GraphMVP pre-trains GNNs on another molecular dataset with 3D geometry. To further support the above claims, we plot the training and testing accuracy curves in Appendix G.

**Observation III:** As demonstrated in Table 2, MAM can serve as a fundamental pre-training sub-task like AttrMask. Furthermore, MAM shows significant superiority over AttrMask when they serve as sub-tasks for multi-task GNNs pre-training.

**Observation IV (Ablation Study):** We substitute or remove some components of the proposed approaches to study their effectiveness. As can be observed in Table 1, the group VQ-VAE is superior to vanilla VQ-VAE in MAM because it prevents the atoms of different types be allocated with the same token id. Additionally, we observe a notable performance drop when we remove the triplet loss $\mathcal{L}_{\text{tri}}$ or contrastive loss $\mathcal{L}_{\text{con}}$ in TMCL, which indicates both of them are necessary and effective.

Table 2: Performance of AttrMask and MAM when they serve as fundamental pre-training sub-tasks. The data for supervised pre-training (Hu et al., 2020) comes from a preprocessed ChEMBL dataset (Gaulton et al., 2012) with some labels from biochemical assays.

| | Tox21 | ToxCast | Sider | ClinTox | MUV | HIV | BBBP | Bace | Average |
|---|---|---|---|---|---|---|---|---|---|
| MGSSL (AttrMask) | 75.2 (0.6) | 63.3 (0.5) | 61.6 (1.0) | 77.1 (4.5) | 77.6 (0.4) | 75.8 (0.4) | 68.8 (0.6) | 78.8 (0.9) | 72.28 |
| **MGSSL (MAM)** | 76.6 (0.7) | 64.5 (0.9) | 62.1 (0.8) | 78.2 (3.8) | 78.7 (0.5) | 76.9 (0.7) | 70.5 (1.1) | 80.2 (1.5) | **73.46** |
| Supervised (Hu et al., 2020) | 76.8 (0.8) | 65.2 (0.5) | 61.7 (0.8) | 57.0 (2.8) | 79.8 (1.6) | 74.3 (1.5) | 67.9 (0.9) | 77.7 (0.8) | 70.05 |
| Supervised + AttrMask | 77.8 (0.6) | 65.3 (0.8) | 63.2 (0.8) | 73.8 (3.6) | 80.9 (1.6) | 77.5 (1.3) | 66.8 (1.4) | 80.7 (1.3) | 73.25 |
| **Supervised + MAM** | 78.6 (0.5) | 66.9 (0.4) | 64.0 (1.0) | 75.4 (2.9) | 81.8 (1.6) | 78.8 (1.0) | 69.1 (1.7) | 82.3 (1.2) | **74.61** |

**Influence of GNNs Backbone.** As shown in Table 3, we verify that Mole-BERT is agnostic to the GNN architectures by trying four popular GNN models including GIN (Xu et al., 2019), GCN (Kipf & Welling, 2017), R-GCN (Schlichtkrull et al., 2018) and GraphSAGE (Hamilton et al., 2017). As can be observed, Mole-BERT achieves consistent and notable improvements over training from scratch with various GNNs. Additionally, pre-training with GIN achieves the most significant gains.

Table 3: Compare pre-training gains (averaged ROC-AUC (%) on 8 datasets) with different GNN architectures. The relative gains mean the relative improvements of Mole-BERT over 'No pretrain'.

| Model | GCN | GIN | R-GCN | GraphSAGE |
|---|---|---|---|---|
| No pretrain | 68.77 | 67.15 | 68.32 | 68.46 |
| MAM | 71.35 | 72.16 | 70.76 | 71.55 |
| TMCL | 68.93 | 71.24 | 69.25 | 69.58 |
| Mole-BERT | 73.22 | 74.04 | 73.51 | 73.74 |
| Relative gain | +6.47 % | +10.26 % | +7.60 % | +7.71 % |

**Broader Range of Downstream Tasks.** We report the performance in regressive property prediction and Drug-target affinity (DTA) tasks in Table 4. DTA is a crucial task in drug discovery, where we aim to predict the affinity scores between the molecular drugs and protein targets. We follow the settings of a recent work (Nguyen et al., 2021) on DTA which models the molecular graphs with GNN and target protein (as an amino-acid sequence) with a convolution neural network (CNN). We substitute the GNN in their approach with pre-trained GNNs. The superior performance indicates that Mole-BERT can work well in a broader range of downstream tasks.

Table 4: Results for molecular property prediction (regression) and DTA (regression). We report the mean (and standard variance) RMSE of 3 seeds with scaffold splitting for molecular property prediction, and the mean (and standard variance) MSE for 3 seeds with random splitting on DTA tasks. *Both indicators are the less the better.* The best result for each task is highlighted in **bold**.

| Datasets | Molecular Property Prediction (↓) | | | | Drug-Target Affinity (↓) | |
|---|---|---|---|---|---|---|
| | ESOL | Lipo | Malaria | CEP | Davis | KIBA |
| No Pre-train | 1.178 (0.044) | 0.744 (0.007) | 1.127 (0.003) | 1.254 (0.030) | 0.286 (0.006) | 0.206 (0.004) |
| ContextPred | 1.196 (0.037) | 0.702 (0.020) | 1.101 (0.015) | 1.243 (0.025) | 0.279 (0.002) | 0.198 (0.004) |
| JOAO | 1.120 (0.019) | 0.708 (0.007) | 1.145 (0.010) | 1.293 (0.003) | 0.281 (0.004) | 0.196 (0.005) |
| GraphMVP | 1.064 (0.045) | 0.691 (0.013) | 1.106 (0.013) | **1.228** (0.001) | 0.274 (0.002) | 0.175 (0.001) |
| AttrMask | 1.112 (0.048) | 0.730 (0.004) | 1.119 (0.014) | 1.256 (0.000) | 0.291 (0.007) | 0.203 (0.003) |
| **MAM** | 1.098 (0.025) | 0.711 (0.010) | 1.107 (0.009) | 1.240 (0.006) | 0.278 (0.005) | 0.188 (0.007) |
| **TMCL** | 1.116 (0.042) | 0.704 (0.014) | 1.123 (0.017) | 1.262 (0.011) | 0.282 (0.005) | 0.194 (0.002) |
| **Mole-BERT** | **1.015** (0.030) | **0.676** (0.017) | **1.074** (0.009) | 1.232 (0.009) | **0.266** (0.004) | **0.157** (0.001) |

Table 5: The performance with various vocabulary sizes on 8 MoleculeNet datasets.

| Vocabulary size | 128 | 256 | 512 | 1,024 | 2,048 |
|---|---|---|---|---|---|
| MAM | 71.42 | 71.65 | 72.16 | **72.21** | 71.66 |
| Mole-BERT | 73.23 | 73.56 | **74.04** | 74.02 | 73.86 |

**Influence of the Vocabulary Size.** In our previous experiments, we set the vocabulary size as 512. However, we cannot decide the optimal vocabulary size for MAM because the exact atoms sub-types are unavailable. In Table 5, we study the vocabulary size ranging from 128 to 2,048, from which we can observe: **(1)** Even if when we set the vocabulary size as 128, near 119 of AttrMask, MAM can outperform AttrMask, which indicates that the tokens derived by VQ-VAE are context-aware and is superior to pure atoms' identities; **(2)** Vocabulary size also influence on MAM's performance. Although the vocabulary size of 1,024 outperforms 512, the superiority is not significant. Hence, we set 512 as the default vocabulary size considering the computational budget.

Figure 5: The query molecule and 5 closest molecules with the extracted representations.

**Molecule Retrieval.** For more comprehensive evaluations, we first extract the representation for a query molecule. And then, we calculate its cosine similarities with all reference molecules in ToxCast dataset. We demonstrate 5 molecules that are most similar to the query molecule with the cosine similarities in Figure 5. As can be observed, the representation similarities of Mole-BERT are approximately aligned with the fingerprint similarities, which indicates that Mole-BERT learns chemically meaningful representations. Moreover, the representations extracted from MAM fail to model the varying degree of similarities between molecules. Hence, graph-level tasks like TMCL are necessary and effective for molecule retrieval. More ablations and results can be seen in Appendix D.

## 6 CONCLUSIONS AND FUTURE WORKS.

In this paper, we find the negative transfer issue of AttrMask can be attributed to the extremely small and unbalanced atom vocabulary. As a remedy, we contribute a context-aware tokenizer with group VQ-VAE. With the new vocabulary, we propose a more suitable pre-training task, MAM, to mitigate the negative transfer issue of AttrMask. Additionally, we develop triplet masked contrastive learning (TMCL) to model the varying degree of molecular similarities. MAM and TMCL constitute a joint pre-training framework (Mole-BERT), which achieves superior performance over state-of-the-art methods while not requiring any domain knowledge. For the future, it remains to be explored whether the proposed pre-training strategies can be applied to protein science (Hu et al., 2022; Tan et al., 2022), where small and unbalanced vocabulary could also impair performance.

# 7 ACKNOWLEDGEMENTS

We thank the anonymous reviewers for their constructive and helpful reviews. This work was supported by the National Key R&D Program of China (Project 2022ZD0115100), the National Natural Science Foundation of China (Project U21A20427), the Research Center for Industries of the Future (Project WU2022C043), and the Competitive Research Fund (Project WU2022A009) from the Westlake Center for Synthetic Biology and Integrated Bioengineering.

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

# A  DATASETS

In this section, we provide detailed information of the datasets for molecular property prediction (classification and regression) and drug target affinity prediction in Table 6. Kindly note the labels for molecular property prediction are scarce because molecular labeling is often expensive (Xia et al., 2022a). More information on these datasets can be found in GraphMVP (Liu et al., 2022a).

Table 6: Summary for the molecule datasets for downstream tasks.

| Dataset | Task | # Tasks | # Molecules | # Proteins | # Molecule-Protein |
|---------|------|---------|-------------|------------|---------------------|
| BBBP | Classification | 1 | 2,039 | – | – |
| Tox21 | Classification | 12 | 7,831 | – | – |
| ToxCast | Classification | 617 | 8,576 | – | – |
| Sider | Classification | 27 | 1,427 | – | – |
| ClinTox | Classification | 2 | 1,478 | – | – |
| MUV | Classification | 17 | 93,087 | – | – |
| HIV | Classification | 1 | 41,127 | – | – |
| Bace | Classification | 1 | 1,513 | – | – |
| Delaney | Regression | 1 | 1,128 | – | – |
| Lipo | Regression | 1 | 4,200 | – | – |
| Malaria | Regression | 1 | 9,999 | – | – |
| CEP | Regression | 1 | 29,978 | – | – |
| Davis | Regression | 1 | 68 | 379 | 30,056 |
| KIBA | Regression | 1 | 2,068 | 229 | 118,254 |

**Input graph representation.** For simplicity, we use a minimal set of node and bond features that unambiguously describe the two-dimensional structure of molecules following previous works (Hu et al., 2020). We use RDKit (Landrum et al., 2013) to obtain these features.

- Node features:
  - Atom number: $1 \sim 118$
  - Chirality tag: {unspecified, tetrahedralcw, tetrahedralccw, other}
- Edge features:
  - Bond type: {single, double, triple, aromatic}
  - Bond direction: {−, endupright, enddownright}

# B  MORE EXPERIMENTAL DETAILS

As we describe in the main text, we use the GIN architecture as the main encoder following previous works (Hu et al., 2020), which make some minor modifications to include bond features. For tokenizer training, we adopt the above 5-layer GINs as the encoder and the decoder, which is trained for 60 epochs on the 2 million unlabeled molecules sampled from the ZINC15 database with the batch size as 256 and the learning rate as 0.001. For TMCL, we set the masking ratios as 0.15 and 0.30, respectively. The temperature parameter $\tau$ is set to 0.1.

# C  THE INFLUENCE OF THE MASKING RATIO

In this section, we study the influence of the masking ratio of MAM and Mole-BERT. As can be observed in Table 7, the performance increase when the masking ratio varies from 0.10 to 0.20 while witnessing a notable drop when the masking ratio varies from 0.20 to 0.30, which indicates over-large masking ratio will impair the GNNs pre-training. Additionally, as shown in Table 8, the masking ratio pairs of two branches in Mole-BERT matter for pre-training. Specifically, when the masking ratio pairs are with larger differences, the performance of Mole-BERT will be pronounced. Also, the performance of Mole-BERT drops sharply when the masking ratio pairs are both larger than 0.25.

Table 7: The performance with various masking ratios of MAM on 8 MoleculeNet datasets.

| Masking ratio | 0.10 | 0.15 | 0.20 | 0.25 | 0.30 |
|---|---|---|---|---|---|
| MAM | 71.64 | 72.16 | 72.21 | 71.51 | 71.36 |

Table 8: The performance with various masking ratio pairs of Mole-BERT on 8 MoleculeNet datasets.

| Masking ratio pairs | (0.15, 0.20) | (0.15, 0.25) | (0.15, 0.30) | (0.20, 0.25) | (0.20, 0.30) | (0.25, 0.30) |
|---|---|---|---|---|---|---|
| Mole-BERT | 73.02 | 73.81 | 74.04 | 72.95 | 73.72 | 72.33 |

## D    MORE RESULTS OF MOLECULE RETRIEVAL

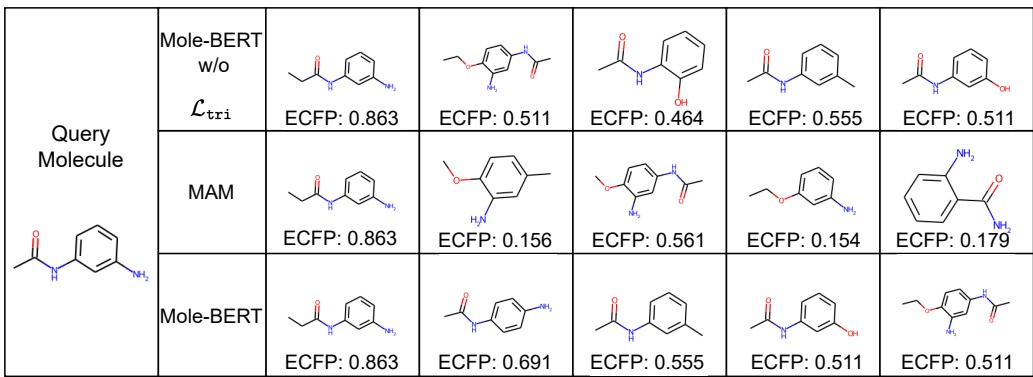

Figure 6: The query molecule and 5 closest molecules with the extracted representations.

We show more results of molecule retrieval in Figure 6, where 'Mole-BERT w/o $\mathcal{L}_{\text{tri}}$' means that we remove the triplet loss of Mole-BERT. As can be observed, the retrieval results of MAM are unsatisfactory. In contrast, both 'Mole-BERT w/o $\mathcal{L}_{\text{tri}}$' and 'Mole-BERT' can find chemically similar molecules in the database. However, 'Mole-BERT w/o $\mathcal{L}_{\text{tri}}$' fails to sort the molecules in a chemically meaningful order. Hence, the triplet loss is necessary and effective for Mole-BERT.

## E    IMPLEMENTATION DETAILS OF BASELINES

Considering that previous works adopt different evaluation protocols, we reproduce all the results with the same protocol as the pioneering work (Hu et al., 2020) rigorously for fairness. Specifically, we fine-tune the respective publicly available pre-trained models with 10 random seeds (0-9) and scaffold splitting. We use a batch size of 32 and a dropout rate of 0.5. We train models on each dataset for 100 epochs and report the test performance when the optimal validation performance is achieved, instead of the results of the last epoch like (Xu et al., 2021b; Hou et al., 2022). Additionally, we evaluate test performance on downstream tasks using ROC-AUC with the validation early stopping protocol, i.e., test ROC-AUC at the best validation epoch is reported. For datasets with multiple prediction tasks, we take the average ROC-AUC across all their tasks. For the recent state-of-the-art method GraphMVP (Liu et al., 2022a), we adopt the contrastive variant (GraphMVP-C). Specifically, we pre-train the model from scratch with the default settings in their paper and fine-tune the pre-trained model with the above-mentioned evaluation protocols.

## F    DISCRETE VAE V.S. GROUP VQ-VAE

Both discrete VAE (Ramesh et al., 2021) and our group VQ-VAE can tokenize the atoms to compact codes. In Table 9, we compare their performance, from which we can observe that our group VQ-VAE outperforms discrete VAE by a significant margin. The reason is that group VQ-VAE can tokenize

Table 9: The comparisons between discrete vae and group VQ-VAE.

| Pre-training Tasks | AttrMask | MAM (Discrete VAE) | MAM (Group VQ-VAE) |
|---|---|---|---|
| Performance | 70.81 | 71.48 | 72.16 |

the atoms in a context-aware and semantics-aware manner. Also, group VQ-VAE can prevent the atoms of different types from being allocated with the same token id.

## G  TRAINING AND VALIDATION CURVES

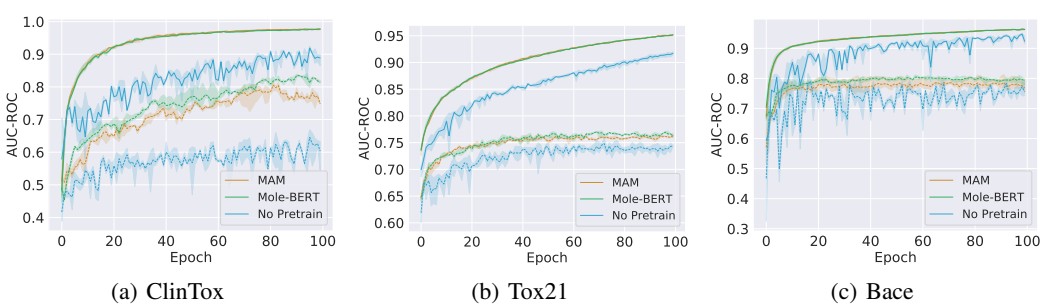

(a) ClinTox  (b) Tox21  (c) Bace

Figure 7: Training (solid lines) and Validation (dashed lines) curves of various pre-training methods on ClinTox, Tox21, and Bace datasets.

We plot the training and validation curves in Figure 7, from which we can observe that the pre-trained models outperform training from scratch by significant margins. Additionally, for small-scale datasets such as Bace, training from scratch tends to overfit the training data of downstream tasks (Xia et al., 2022d). In contrast, the pre-trained models can mitigate the over-fitting issue.

## H  ABLATIONS FOR GROUP VQ-VAE.

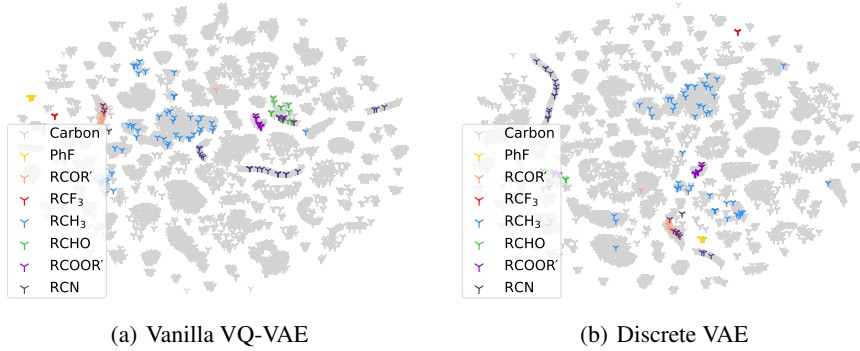

(a) Vanilla VQ-VAE  (b) Discrete VAE

Figure 8: The t-SNE visualizations of the carbon representations learned by vanilla VQ-VAE and discrete VAE (Ramesh et al., 2021).

In this section, we substitute group VQ-VAE with vanilla VQ-VAE or discrete VAE (Ramesh et al., 2021) and show the t-SNE visualization of the carbon representations learned by them. As can be observed in Figure 8, although vanilla VQ-VAE and discrete VAE can distinguish the carbons well based on the types of functional groups that carbons belong to, they cannot prevent some atoms of different types from being clustered in the same region. In contrast, our group VQ-VAE can alleviate this issue.

# I    DETAILED INFORMATION OF THE STRUCTURAL ABBREVIATIONS.

Table 10 shows the seven local structure categories of the carbons visualized in the main text. The codes and datasets for visualization are from a recent work (Wang et al., 2023).

Table 10: The first column lists the structural abbreviations corresponding to the legends. The second column list the corresponding chemical groups. The third column shows the structural formula.

| Abbr | Name | Structural Formula |
|------|------|--------------------|
| RPhF | Fluorophenyl | R—〈benzene ring〉*—F |
| $RCF_3$ | Trifluoromethyl | $R-\overset{\overset{F}{\vert}}{\underset{\underset{F}{\vert}}{C}}-F$ |
| RCHO | Aldehyde | $R-\overset{\overset{O}{\Vert}}{C}-H$ |
| RCOOR' | Ester | $R-\overset{\overset{O}{\Vert}}{C}-OR'$ |
| RCOR' | Ketone | $R-\overset{\overset{O}{\Vert}}{C}-R'$ |
| RCN | Nitrile | $R-C\equiv N$ |
| $RCH_3$ | Methyl | $R-\overset{\overset{H}{\vert}}{\underset{\underset{H}{\vert}}{C}}-H$ |

