# OpenReview forum: "Mole-BERT: Rethinking Pre-training Graph Neural Networks for Molecules"
_ICLR.cc/2023/Conference — ICLR 2023 poster_

### Official Review · Reviewer_E4Mj · 2022-10-23

**Confidence:** 3
**Correctness:** 3
**Technical Novelty And Significance:** 3
**Empirical Novelty And Significance:** 3
**Recommendation:** 6

**Clarity, Quality, Novelty And Reproducibility:**

This paper is well written and organized. Quality and novelty are good. No code is provided.

**Strength And Weaknesses:**

Strength:

- Existing masked attribute pretraining task is too easy for GNN to learn more meaningful atom representation and motif or context prediction is not as comprehensive as the proposed tokenization method. The proposed tokenizer is simple and effective. Visualization in Figure 1 illustrates the effectiveness of the proposed method.

- A new graph-level contrastive pretraining method is presented. A triplet consists of differently masked (smaller masked ratio and larger masked ratio) graphs of the same molecule, and the contrastive loss is defined for the triplet.


Weaknesses:

- Unreliable results in Table 1. Reported results of GraphMAE in Table is different from the original paper. GraphMAE paper reports 73.8% average ROC-AUC while Table 1 shows 71.33%. Could the authors clarify the resources of reported numbers in Table 2 of existing methods?

- Missing related work reference. Could the authors include the pioneering work [1] which first applies masked pretraining for molecule data?


[1] Wang, Sheng, et al. "SMILES-BERT: large scale unsupervised pre-training for molecular property prediction." Proceedings of the 10th ACM international conference on bioinformatics, computational biology and health informatics. 2019.

**Summary Of The Paper:**


This paper proposes a novel node-level pretraining task and a graph-level contrastive learning task for molecule graph representation learning. The node-level pretraining task is to predict a discrete tokenized atom that also encodes the context information of the target masked atom. This new task is more challenging and helps the GNN learn meaningful representation. The graph-level pretraining method differentiate differently masked graphs of the same molecule. The proposed pretraining method is effective and can match or outperform state-of-the-art methods on multiple downstream tasks.


**Summary Of The Review:**

The authors study the GNN pretraining for molecule data and proposed novel node-level and graph-level pretraining methods. Extensive experiments show the effectiveness. However, there are some critical concerns about the reported results of baseline methods in comparison.

---

### Official Review · Reviewer_nohC · 2022-10-25

**Confidence:** 3
**Correctness:** 3
**Technical Novelty And Significance:** 3
**Empirical Novelty And Significance:** 3
**Recommendation:** 8

**Clarity, Quality, Novelty And Reproducibility:**

### Novelty

- Incorporating VQVAE and using the latent codes as the targets for masked prediction is novel and brilliant.
- Group VQVAE is a good hack.
- Although the contrastive loss w.r.t. to different masked versions & the triplet loss is already known to the community, showing that they can be combined with the MAM objective and improve performance is also interesting.

### Clarity

The paper is well-written. The description and motivation for each algorithmic component are clear. The logic of the entire paper is clear and well-structured.

### Reproducibility

The paper includes adequate details of the experiments. The authors claim that they will release the code.

**Strength And Weaknesses:**


Strengths:

- The authors identify the reason of failure of AttrMask, which is interesting and also intuitive: (1) the atom attribute is easy to predict from its neighbours (2) the atoms are extremely unbalanced. Therefore, masked prediction task is too easy and fine-grain for molecule pretraining.

- The idea of using VQ-VAE to get contextual codes of atoms is brilliant and interesting. The group VQ-VAE modification is also clever.

- Contrastive learning with adding random masks being the only data augmentation naturally fits into the masked pretraining framework. The triplet loss is also well-motivated.

- The experimental results are strong. The authors managed to reproduce all 17 previous results and demonstrated the superiority of their techniques.

- The ablation studies are thorough and informative. The authors show the necessities of all the algorithmic components: contrastive loss, triplet loss, and VQ-VAE-based masked pretraining. Furthermore, the authors conduct experiments on the masked prediction objective (classification v.s. regression), tSNE visualization of the latent codebooks, and etc.

Other comments:

- For GraphMVP (Liu et al., 2022a), the reported performance is 72.64. While in the original paper the number is 73.07 for GraphMVP-C.

- Eq (8) should have a minus sign, since you are maximizing the similarity between $h_G$ and $h_{M_1}$.

**Summary Of The Paper:**


The paper how to  pretrain GNNs for molecules. The authors identify that the issue of the previous masked prediction method (AttrMask) is due to small and unbalanced atom vocabulary, which makes the task easy and hinders the GNNs from learning useful information. Then the paper proposes a new a approach: (1) first the authors train a VQ-VAE for molecules, whose latent space contains context-aware, semantically-rich discrete codes for the atoms; (2) next the authors train another GNN to predict the latent codes from masked molecules. Besides the improvement to mask prediction pretraining, the authors further improve upon the contrastive learning objective by adding a triplet loss as a regularizer, which encourages the graph-level embeddings of the original molecule and the 15% masked version to be closer than the embeddings of the  original molecule and the 30% masked version. By combining all the tricks, the authors are able to show strong performance on a set of molecular property prediction tasks. The authors also conduct adequate ablation studies to validate the effectiveness of all the algorithmic components.


**Summary Of The Review:**

Good motivations, brilliant ideas, and strong experimental results.

---

### Official Review · Reviewer_9dQ7 · 2022-10-26

**Confidence:** 4
**Correctness:** 3
**Technical Novelty And Significance:** 3
**Empirical Novelty And Significance:** 3
**Recommendation:** 6

**Clarity, Quality, Novelty And Reproducibility:**

I have some concerns/questions, mainly regarding presentation and empirical evaluation:
- The connection between MAM and Denoising VAE is unclear to me. Is the graph masking (during MAM training) influencing the latent codes of VQ-VAE? or is VQ-VAE trained in a separate stage and kept fixed?
- I would say the connection with D-VAE seems quite isolated from the rest of the paper. Also, I am not sure how much this connection adds to it. I believe this connection should be clarified/better formalized or moved to Appendix.
- It would be helpful to include the same visualization in Fig. 1(c) for other baselines in the Appendix.
- What are the frequencies of learned atom types obtained from the proposed tokenizer (i.e., VQ-VAE codes) --- similarly to Fig. 1(b)?
- It seems that the paper follows the evaluation setup by Hu et al., 2020. But there are some significant differences between the results reported in this paper and those by Hu et al., 2020. For instance, 76.7+-0.4 vs. 75.1+-0.9 on Tox21. Why is this the case?
- The paper claims that MAM (with vocab. size 128) (71.42) in Table 5 is significantly better than AttrMask (70.81). Is it true if we consider the standard deviation?
- The group size for each atom type seems arbitrary, and the paper does not discuss the impact of such a choice. For instance, what would be the impact of dividing codebook groups according to the atom distribution?
- Minor issue: Fig (3) may be misleading as it shows 3 GNNs.

The novelty mainly comes from the context-aware tokenizer and triplet loss term, which account for varying degrees of similarity between molecules. Overall, the proposed approach is somewhat novel while some aspects of it are available in previous works.

Regarding reproducibility, the code is not available during the reviewing phase. The authors provide implementation details in the main paper and the Appendix. However, importantly, the paper has re-run experiments for the baselines, but these implementation details have not been reported.

**Strength And Weaknesses:**

Strengths
- The paper conducts comprehensive experiments w/ recently introduced methods, including ablation studies on GNN architectures, vocabulary size, and model hyper-parameters.
- The idea is simple and intuitive. Also, the paper reads well.

Weaknesses
- The motivation behind some modeling choices seems a bit hand-wavy and not theoretically grounded. For instance, the paper justifies the use of VQ-VAE mainly due to the success of DALL-E.
- Overall, it is not clear if the proposed method yields significant improvements considering the standard deviation of the accuracy metrics. Some recent works have raised similar issues (e.g., https://arxiv.org/pdf/2207.06010.pdf).


**Summary Of The Paper:**

The paper introduces a new framework for pre-trained representations of molecules. It first discusses issues related to masked language modeling approaches for molecular graphs (AttrMask), e.g., negative transfer. To overcome these issues, the paper proposes using a VQ-VAE-based tokenizer to leverage context-dependent node representations followed by a joint node-level (called MAM) and graph-level (called TMLC) pre-training strategy. The whole framework is called Mole-BERT. Experiments on widely adopted benchmarks for downstream molecular property prediction show the efficacy of Mole-BERT.


**Summary Of The Review:**

I have some concerns about the significance of performance gains from Mole-BERT over existing baselines. Since the paper somehow aims to bridge the gap between masked language pre-training in text and molecules, I would expect stronger gains. There are also points in the paper to be clarified and better discussed. However, I overall like the idea of context-aware tokenizers for molecular data, and I think this is a promising direction. Therefore, I am slightly inclined toward accepting the paper.

---

### Official Review · Reviewer_4oZ7 · 2022-10-31

**Confidence:** 4
**Correctness:** 3
**Technical Novelty And Significance:** 2
**Empirical Novelty And Significance:** 2
**Recommendation:** 6

**Clarity, Quality, Novelty And Reproducibility:**

The employed baselines were not introduced clearly. Some baselines are for general graphs. It is unclear how they were pre-trained by what datasets. GraphMVP is a pre-trained model on another molecular dataset with 3D geometry. Does Mole-BERT use much more larger dataset for pre-training than GraphMVP? is the 1.34% improvement due to the larger  dataset for pre-training in Mole-BERT ?

There are other molecular pre-training models, such as GROVER (Rong et al., 2020a)  and (Liu et al., 2022a; Sta ̈rk et al., 2021; Fang et al., 2022a; Zhu et al., 2022).  They should be included as baseline for comparison as well.

Another question about the results in Table 1 is the significance of performance improvement. The proposed Mole-BERT mostly outperforms the best baseline by around 1%. However, the std is also around 1.0. It is unknown how statistically significant the improvement is.


**Strength And Weaknesses:**

The strengths of the paper are:
1) authors conducted extensive evaluation experiments on several datasets.  The pre-trained model was evaluated for molecular property prediction and drug-target affinity prediction.
2) the experimental results show that the proposed pre-trained model has better performance than the baselines.

The weaknesses of the paper are:
1) the proposed pre-training framework is a mixture of existing tools. The variant of VQ-VAE  is only a small change of VQ-VAE by organizing atoms in groups in the codebook (assigning fixed intervals to C, N O and other atoms). The masked modeling is a well developed idea, as well as the graph-level contrastive learning.
2) These existing tools are employed without any adaptation to the molecular graphs. For example, the MAM is used just with randomly mask its 15% atoms’ tokens, and the graph augmentation in graph-level contrastive learning is just based on masking 15% vs masking 30%. Molecular graphs are different from other general graphs, because of their constraints on the graph validity. These  constraints should be considered in the design of self-supervising tasks.
3) the paper has writing errors such as  "an one"

**Summary Of The Paper:**

This paper presents a pre-training framework named Mole-BERT for learning molecular representation. The key components of Mole-BERT are 1) a variant of VQ-VAE for getting discrete tokenization result of atoms in molecular graphs, 2) a Masked Atoms Modeling strategy for predicting the masked atoms, and 3)  a graph-level contrastive learning task for adjusting the molecular representations. Authors evaluated Mole-BERT on molecular property prediction and drug-target affinity prediction, and compared with a number of baselines. Ablation study was conducted as well for showing the effectiveness of each component in the designed framework.

**Summary Of The Review:**

In general, it is an interesting paper. However, the proposed model is weak on technique novelty. It is more like a model with mixed existing tools.  The experimental evaluation misses details about the baselines, and cannot support strongly the effectiveness of the proposed model.

---

### Decision · Program_Chairs · 2023-01-20

**Decision:**

Accept: poster

**Justification For Why Not Higher Score:**

The technical novelty is not super.

**Justification For Why Not Lower Score:**

The overall idea makes sense and worth sharing to the community.

**Metareview: Summary, Strengths And Weaknesses:**

This paper proposes Mole-BERT, a two-stage GNN pre-training method for molecules. At the first stage, the paper presents a novel Group VQ-VAE as a context-aware tokenizer of given atoms that mitigate the quantitative divergence between dominant and rare atoms. With discrete tokens learned from Group VQ-VAE, the method introduces two pre-training objectives for the second-stage pre-training: Masked Atoms Modeling (MAM) and Triplet Masked Contrastive Learning (TMCL).  Here, the paper designs MAM to predict encoded discrete values for each masked atom, and designs TMCL as a triplet loss to reflect the heterogeneous similarity between the anchor and other molecules.

After the discussion phase, the reviewers unanimously ended up supporting the acceptance of this paper, and the authors were quite responsive in addressing the reviewer's concerns. Overall, the reviewers appreciated its strong empirical results with informative and thorough ablation studies. AC also agrees with that and thinks the paper presents a promising direction for molecular representation learning. Overall, AC recommends acceptance.

**Note From Pc:**

if the above contains the word "oral" or "spotlight" please see: "oral" presentation means -> notable-top-5% and "spotlight" means -> notable-top-25%. As stated in our emails, we are disassociating presentation type from AC recommendations